# “If It Works in People, Why Not Animals?”: A Qualitative Investigation of Antibiotic Use in Smallholder Livestock Settings in Rural West Bengal, India

**DOI:** 10.3390/antibiotics10121433

**Published:** 2021-11-23

**Authors:** Jean-Christophe Arnold, Dominic Day, Mathew Hennessey, Pablo Alarcon, Meenakshi Gautham, Indranil Samanta, Ana Mateus

**Affiliations:** 1Veterinary Epidemiology, Economics and Public Health Group, Royal Veterinary College, Hawkshead Lane, Hatfield AL9 7TA, UK; mphennessey@rvc.ac.uk (M.H.); palarcon@rvc.ac.uk (P.A.); 2South Wales Farm Vets, Tynewydd Farm, Cardiff CF72 8NE, UK; 3Department of Global Health and Development, Faculty of Public Health and Policy, London School of Hygiene & Tropical Medicine, 15-17, Tavistock Place, London WC1H 9SH, UK; Meenakshi.Gautham@lshtm.ac.uk; 4Department of Veterinary Microbiology, West Bengal University of Animal and Fishery Sciences, Kolkata 700037, India; isamanta76@gmail.com; 5World Organisation of Animal Health (OIE), 12 Rue de Prony, 75017 Paris, France; a.mateus@oie.int

**Keywords:** antibiotic usage, antibiotic resistance, smallholder, livestock, poultry, animal health, antibiotic stewardship, qualitative, India, One Health

## Abstract

Smallholder farms are the predominant livestock system in India. Animals are often kept in close contact with household members, and access to veterinary services is limited. However, limited research exists on how antibiotics are used in smallholder livestock in India. We investigated antibiotic supply, usage, and their drivers in smallholder livestock production systems, including crossover-use of human and veterinary antibiotics in two rural sites in West Bengal. Qualitative interviews were conducted with key informants (n = 9), livestock keepers (n = 37), and formal and informal antibiotic providers from veterinary and human health sectors (n = 26). Data were analysed thematically and interpreted following a One Health approach. Livestock keepers and providers used antibiotics predominantly for treating individual animals, and for disease prevention in poultry but not for growth promotion. All providers used (highest priority) critically important antimicrobials for human health and engaged in crossover-use of human antibiotic formulations in livestock. Inadequate access to veterinary drugs and services, and a perceived efficacy and ease of dosing of human antibiotics in animals drove crossover-use. Veterinary antibiotics were not used for human health due to their perceived adverse effects. Given the extent of usage of protected antibiotics and crossover-use, interventions at the community level should adopt a One Health approach that considers all antibiotic providers and livestock keepers and prioritizes the development of evidence-based guidelines to promote responsible use of antibiotics in animals.

## 1. Introduction

Antibiotics are widely used in food-producing animals to prevent and treat infections and for growth promotion. This has resulted in the development and spread of antibiotic resistance (ABR) in animal populations [1]. ABR is a pressing concern within the agricultural sector; not only for its repercussions for animal health, welfare, and food production, but also due to the risk of the spread of ABR to humans [2,3]. This is believed to occur through direct contact, via the food chain, or via the environment; however, the extent to which ABR in humans is attributable to animals and derived food products is yet to be determined [2,4,5]. Recognising the cross-species and environmental spread of ABR, the World Health Organisation (WHO), the Food and Agriculture Organisation of the United Nations (FAO), and the World Organisation for Animal Health (OIE) have created the Tripartite initiative that also now includes the United Nations Environment Programme (UNEP) [2]. The Tripartite recommends that a One Health, inter-sectorial approach be adopted to mitigate ABR. 

Global ABU in food animals is expected to rise by 67% between 2010 and 2030, from an estimated 63,151 (±1560) tonnes to 105,596 (±3605) tonnes [4]. Currently India is one of the highest consumers of antibiotics in food-producing animals worldwide, with their consumption expected to double by 2030 [4,6]. However, details of how and for what purposes these antibiotics are used in food-producing animals are limited.

In commercial livestock systems in India, antibiotics are reported to be widely used for disease prevention and growth promotion which are established drivers of ABR [7,8]. Rural smallholder livestock systems predominate in India, yet few studies have investigated how antibiotics are used in these settings [7,9]. 

Here, animals are often in closer contact with household members, with potentially greater risk of exchanging pathogens and antibiotic resistance genes across species [10]. Inadequacies in rural veterinary health provision and access may also drive the misuse of antibiotics in rural, smallholder production settings [9].

Public provision of veterinary health care is universally inadequate for the rural livestock population, with an estimated one qualified veterinarian available per 10–15 villages in India [9]. Limited access to formal veterinary expertise and the perceived high cost of veterinary services leaves smallholders heavily reliant on the use of antibiotics and the services of a diverse cadre of veterinary paraprofessionals (VPPS) [11,12,13,14]. VPPs are important primary sources of livestock disease management despite having a range of durations and formalities of training [9,13,15]. Although antibiotic provision is outside of their remit, VPPs have been found to irrationally use antibiotics in similar Indian settings [13,15]. 

In India, farmers have access to a pluralistic healthcare system, whereby several systems of medicine are practiced through both formal and informal channels [13,15,16]. This includes actors from the human sector such as informal providers of human health (IPs), who often have no veterinary training [17]. IPs were prevalent and have been reported to be serving livestock in our study sites [17] and in other comparable settings [18]. Furthermore, farmers are reported to seek antibiotics directly from pharmacies over the counter (OTC) without a veterinary prescription [13,15,18,19,20]. Smallholders also rely on peer advice and their own experiences to manage livestock diseases, often opting to self-treat their animals with antibiotics [13,14]. The utilisation of the human health sector for veterinary needs raises concerns that human antibiotic formulations may be used in livestock in these settings. This is exacerbated by the lack of sufficient veterinary diagnostic capacity in rural India, which further reduces the ability of animal health providers to make informed antibiotic selection decisions [13]. 

Although not previously identified in India, “crossover-use” of human and veterinary antibiotics has been reported in comparable low-to-middle income settings, including in Guatemala, Uganda, Nigeria, and Cambodia [21,22,23,24,25,26]. This phenomenon refers to instances whereby antibiotics intended for human usage are used in animals and vice versa [21,22,23,24,25,26].

This is concerning primarily due to the potential of resistance developing to classes of antibiotics used in human health within animal populations and its onward spread into human populations [2,5,27]. Furthermore, there is little information on the safety and efficacy of human formulations in animals, or animal formulations in humans, and so their cross-over usage may be ineffective or toxic for the recipient [28,29]. No study to date has aimed to investigate crossover-use specifically, but rather observed this phenomenon during investigations into ABU more broadly. Therefore, the current literature lacks insight into drivers of this practice; however, concerns for the crossover-use of antibiotics in rural smallholder settings are merited and should be investigated due to its implications for public health [2,10,30]. 

To develop effective antibiotic stewardship interventions, a more comprehensive understanding of ABU practices in smallholder settings in rural India is needed. This study aims to qualitatively investigate the provision of animal healthcare and ABU in smallholder livestock systems, the drivers of ABU, and the crossover-use of veterinary and human antibiotic formulations in humans and livestock in two village clusters of rural West Bengal, India.

## 2. Results

### 2.1. Descriptive Analysis

Across the two study sites, a total of 67 participants were interviewed (Table 1); nine key-informants, 26 antibiotic providers (five of whom were also key-informants), and 37 livestock keeping households—23 in Site 1 and 14 in Site 2.

#### 2.1.1. Livestock Keeping Practices

In both sites, cattle, goats, and poultry were commonly reared in participating households. Livestock were used both as sources of food and income; their produce (meat, milk, eggs) was utilised for household consumption, or sold locally to supplement household income. In Site 1, only two small-scale commercial poultry enterprises were identified, whilst more commercial production systems were identified in Site 2 (Appendix A). Interestingly, this was despite Site 2 being the more remote of the two sites, accessible only via a ferry across a river. However, a local hospitality sector was described that catered to tourists in the area (the Sunderbans) that found it cheaper and more practical to produce poultry locally than to source food animal products from Kolkata.

Livestock, poultry, and their products held socio-cultural significance, to feed visiting guests and during religious and wedding ceremonies. Furthermore, cattle dung was used as a household fuel, and poultry manure was used to fertilise crops. Livestock also provided an economic safety net for their owners for times of financial uncertainty.

Livestock and domestic poultry were housed in close proximity to, or in shared household spaces. Many households were also situated next to small ponds used to raise fish and were also used for bathing and as sources of water for preparing food, household chores, and watering livestock. 

Commercial poultry were housed in purpose-built poultry sheds of varying quality. Most of the commercial poultry producers in Site 2 (four out of seven) used production cycle protocols they had received from a local non-governmental organisation (NGO) engaged in rural development. These contained information on prophylactic ABU, vaccination against “Ranikhet” (Newcastle disease) and “Gumboro” (infectious bursal disease), de-worming, and batch rearing of poultry.

Livestock keepers predominantly learnt how to raise their animals from previous generations of family farmers, or through their own experience. A few described how the local government animal health departments organised animal health camps, offering livestock development advice, treatment, and preventative healthcare. In Site 1, livestock keepers said that these initiatives were infrequent, whilst in Site 2, these were reported to be organised on a regular (monthly) basis.

#### 2.1.2. Livestock Antibiotic Providers

Livestock keepers in both sites had access to animal healthcare, and consequently to antibiotics, through both the public and private animal health sectors, veterinary and human drug shops, and informal providers (IPs) from the human health sector (Table 2).

Treatment of livestock was typically sought when clinical signs of disease were severe or when management of the condition by a livestock keeper was unsuccessful. Failure of ethnoveterinary methods (traditional practices of veterinary medicine) [31], were also described by a smaller proportion of livestock keepers prior to seeking animal healthcare. This included performing particular rituals and the use of indigenous plants. However, the drivers of ethnoveterinary medicine usage in livestock was not within the scope of our study and, therefore, it was not investigated in detail. 

Public veterinary services were quite limited in both sites, more so in Site 1 even though it was closer to urban fringes. Here the only government provider was public VPP known as a Livestock Development Assistant. The nearest qualified veterinarian here was a private one, who lived in a small town 13 km away. By contrast, Site 2 had two government vets, one providing mobile services and the other available on site at the Block Animal Healthcare Centre, approximately 3 km away. A third government veterinarian performed more of an administrative role as the in-charge of this block level centre. 

Other VPPs were active across both sites who served a dual public and private role and are termed public-private VPPs in this paper (Table 2). These were similar across both sites. A distinction between solely public VPPs and those working in a dual public-private capacity is made based on how they were utilised by livestock keepers, and in antibiotic provision. They included Pranibandhus (friends of the animals) and Pranimitras (female friends of the animals) in both sites, and an Animal Development Volunteer (ADV) who was uniquely present in Site 1. In their public capacity, Pranibandhus provided artificial insemination and livestock development services and Pranimitras delivered vaccination to goats and chickens and assisted in the organisation of mobile veterinary camps. The ADV performed a comparable role as the Pranibandhus. All were paid on a commission basis and provided with the materials and resources required to fulfil their public services. In their private capacity, they delivered livestock healthcare informally and were paid directly by livestock keepers. There were practical as well as economic reasons for the provision of private services:


*“If you ask me now how you are practicing, we are only supposed to do vaccination, and artificial insemination. But if we did just that we won’t make enough money, so we, on our own, have learnt how to use antibiotics from other veterinary doctors”.*
Public-private VPP1, Site 1

Purely private sector healthcare was largely provided by a different cadre of VPPs known locally as para-vets. A small number of qualified veterinarians (one for each Site) also operated privately. In both sites, para-vets worked in a similar capacity as Pranibandhus and provided antibiotics to livestock keepers. In Site 2, they had received a two-year training by the local NGO, whilst those in Site 1 had acquired their knowledge from shadowing public and public-private VPPs. Their ongoing training consisted solely of informal interactions with the public VPP when faced with a clinical case that they felt they could not manage. 

Only three retailers of veterinary drugs were identified in this study: two veterinary drug shops situated 13 km outside of Site 1 and a poultry agro-veterinary shop situated in a small town outside of Site 2 approximately 3 km away. Numerous human drug shops were present within or close to both sites. Those near Site 2 stocked some veterinary medicines too, but not those in Site 1. Livestock keepers, public-private VPPs, and para-vets frequented these retail outlets of both types. Informal providers of human health (IPs) were ubiquitous across both sites and occasionally provided livestock treatment and veterinary advice to livestock keepers. 

The major differences between Site 1 and Site 2 regarding their geographical information, livestock production systems present, and availability of veterinary services and access to veterinary medicines are summarised in Table 3.

#### 2.1.3. Antibiotic Use in Livestock

We documented 17 veterinary antibiotic formulations used to treat cattle, goat, and poultry illnesses across both sites (Table 4). Seven of these formulations contained antibiotics classified as highest priority critically important antibiotics (HPCIA) for human medicine, as defined by the WHO [22]. HPCIAs identified included fluoroquinolones (enrofloxacin, marbofloxacin, and ofloxacin), third generation cephalosporins (ceftiofur and ceftriaxone), and macrolides (tylosin). A further five formulations contained antibiotics classified as critically important antibiotics (CIA) for human health. These included penicillins (amoxicillin, ampicillin, ampicillin-cloxacillin), aminoglycosides (gentamicin), and aminoglycoside-cyclic polypeptide combinations (neomycin sulphate-bacitracin). Many of these drugs were used on a first-line basis:


*“In primary cases enrofloxacin or sulfadiazine or oxytetracycline get most of the work done! Then the latest ones like ofloxacin, ceftazidime [third generation cephalosporin] are for quicker action (…). The client that can afford it, we give them a bit more expensive antibiotics such as ceftriaxone, ceftiofur [third generation cephalosporins], ofloxacin (…). If it doesn’t work, we change it to higher [power] antibiotics after three days”.*
Veterinary drug shop 1, Site 1

Antibiotics were administered to livestock either as injectable or oral formulations and were used predominantly to treat individual animals. However, in small scale commercial poultry enterprises in Site 2, enrofloxacin, gentamycin, and oxytetracycline were used prophylactically via medicated water or feed. Injectable antibiotics were obtained from veterinary drug shops and mostly administered by public-private VPPs, para-vets or veterinarians, though occasionally these were administered directly by livestock keepers to their sick animals. There were also instances of self-medication in dairy cattle and goats. 

Antibiotic courses prescribed/dispensed to livestock by the various antibiotic providers were typically for two to three days. Even if longer treatments were prescribed, there could be lack of compliance due to livestock keepers’ lack of knowledge or financial resources:


*“People of [village in Site 1] are mostly poor (…) So, sometime even if the doctor [animal health practitioner] has given [prescribed] medicine for seven days, they would take medicines for two days”.*
Human drug shop 1, Site 1

Treatment failures led to antibiotic providers either extending antibiotic courses, changing to newer or more “powerful” antibiotics, or referring cases to more qualified animal health providers.

#### 2.1.4. Antibiotic Knowledge in Livestock Keepers 

Livestock keepers from both sites demonstrated limited antibiotic knowledge, with no respondent able to explain the term ‘antibiotic’. This was attributed to their self-reported low level of education:


*“We are illiterate people. We can’t read or write. How can we remember the names [of the medicines]?”*
LK1, Site 1

Livestock keepers were often unaware that antibiotics were being administered by providers and would refer to these medicines using generic terms such as “tablet”, “powder”, or “injection”. Livestock keepers trusted their animal healthcare providers to dispense the correct medicines and few providers would explain their treatment plan to them.

#### 2.1.5. Antibiotic Knowledge in Livestock Healthcare Providers 

Public-private VPPs and para-vets in both sites had mixed levels of antibiotic knowledge. They broadly perceived antibiotics as enhancing the ability of the body to fight infections and ABR as the body becoming resistant to the effects of the antibiotic. Some correctly described how ABR presented as a treatment failure subsequent to overusing antibiotics, or through a lack of compliance by clients to recommended treatment regimens. One believed that ABR resulted from poor nutrition:


*“Now the food that’s given to the animals aren’t nutritious enough because everything is made artificially. So, the antibiotics aren’t working anymore. The resistance is growing”.*
Public-private VPP 2, Site 1

Veterinarians had better knowledge of ABR. When asked about their understanding of ABR, their responses included its development through antibiotic overuse and by not completing full courses of antibiotic treatment. However, despite improved awareness, some of the practices of certain veterinarians could be considered inappropriate:


*“We provide antibiotic in viral disease to prevent the secondary bacterial infection”.*
Veterinarian 4, Site 2

Veterinarians, both in the public and private sector, were a key source of knowledge for the para-vets in Site 2, whilst in Site 1, the public-VPP was used.

Human drug shop owners and IPs in both sites had no formal medical or pharmaceutical training, or any veterinary expertise. Human drug shop owners had some knowledge, though their selection of antibiotics for livestock was based on their experience of treating people or mimicking the drug choices of animal healthcare providers.

### 2.2. Drivers of Antibiotic Usage in Livestock

#### 2.2.1. Antibiotics as Therapeutic Treatment

Across both sites, antibiotics were used predominantly for the treatment of both specific diseases (such as mastitis and fowl cholera) and clinical signs (such as lethargy, inappetence, and fever). This was a common finding from livestock keepers, animal health providers, and key informants: 


*“Only when there’s a problem then [people use antibiotics], else not on a regular basis”.*
Key informant 2, Site 1

Most livestock keepers were unaware of any other purpose of ABU (e.g., growth promotion). Instead, improved nutrition, de-worming, vaccination, and vitamin supplementation were their chosen methods for improving livestock productivity.

#### 2.2.2. Antibiotics as Protection against Disease in Poultry

Primarily it was in poultry that antibiotics were used for non-therapeutic purposes. In Site 1, one owner of ten chickens explained that if one of his chickens fell sick, he would treat the whole flock to prevent further spread of infection (i.e., metaphylactic use). In Site 2, three of the seven small-scale commercial poultry enterprises routinely used antibiotics prophylactically (particularly enrofloxacin, a HPCIA) as part of their protocol for rearing broiler birds. 


*“If we follow this voucher [schedule] and give medication, Ranikhet [Newcastle disease] is prevented (…) If we follow that [the schedule], disease don’t come easily”.*
LK26, Site 2

In site 2, the advice to use antibiotics as part of a poultry rearing schedule was provided by public and private veterinarians, and the owner of a poultry shop.

Veterinarians in site 2 and public-private VPPs in site 1 also occasionally prescribed antibiotics prophylactically in cattle affected by systemic viral diseases to prevent secondary bacterial infections:


*“I also try to prevent the diseases. For example, BQ [blackquarter, a vaccine-preventable clostridial disease], FMD [Foot and Mouth Disease, a viral disease] harm the cow’s health a lot. And the treatment is also quite costly. And you have to give antibiotics. So, we give treatment in advance!”*
Public-private VPP 2, Site 1

#### 2.2.3. Lack of Diagnostic Certainty Leads to Heuristics

Selection of antibiotics based on clinical signs was the norm amongst the animal healthcare providers of both sites due to the lack of veterinary diagnostic laboratories. Public-private VPPs, para-vets, drug shop owners, and IPs based their selection of therapeutic course on their knowledge and previous experience of livestock diseases: 


*“In case of cows we measure the temperature using the thermometer in the rectum. And if the cows have cough, or cold, has respiratory infection, we use drugs from the ampicillin group (…) there is no system here to get the blood of the cow tested. It doesn’t happen in this country (…)”.*
Public-private VPP 1, Site 1

Moreover, with many livestock keepers in both sites lacking the means of transporting livestock to the nearest veterinarian, or even to the public-private VPPs and para-vets, it was not always possible for sick livestock to be examined by the animal healthcare providers prior to antibiotic selection. Typically, veterinary and human drug shop owners, public-private VPPs, para-vets, and even IPs would take a verbal history from the livestock keepers and then dispense/prescribe the antibiotics they believed to be most appropriate: 


*“No, I don’t see the animals. I base my diagnosis on what the client is saying. I use my experience to understand what the clients are trying to say and what might have happened. They are not always right (…)”.*
Veterinary drug shop 1, Site 1

In situations where public-private VPPs and para-vets visited households to treat livestock, the antibiotics they brought were based on the clinical history they received by telephone from livestock keepers. Otherwise, the choice of antibiotic would depend on the stock of antibiotics carried by the livestock healthcare provider:


*“Mainly they [healthcare providers] use what they have with them. If they have oxytetracycline, they use it on every animal. They don’t diagnose whether it is bacterial or viral. If he has enrofloxacin he uses enrofloxacin for all animals”.*
Veterinarian 4, Site 2

#### 2.2.4. Antibiotic Usage to Beat Competition and Retain Business

Public-private VPPs and para-vets across both sites reported that they felt the pressure to dispense antibiotics to clients as all their peers did so. They felt that if they were not seen to be providing treatments that worked quickly and efficiently, at a low cost, then clients would seek other providers, resulting in loss of business. The risk of loss of income outweighed the risk of ABR, as explained by one public-private VPP: 


*“With regard to treatment, antibiotics are used more than it was before. Antibiotics should be used as less as possible. But we have to use it still, because of the competition. If we can cure the patient fast, then they would call me. If it takes such a long time, they will want to seek other doctors (…)”.*
Public-private VPP 2, Site 1

#### 2.2.5. Promotion and Incentives by Pharmaceutical Companies

Representatives of veterinary pharmaceutical companies were reported to have a strong community presence across the study sites, promoting their antibiotic products and encouraging ABU. In Site 1, pharmaceutical representatives directly approached livestock keepers to advise on the usage of different antibiotics based on the observed clinical signs in livestock. Livestock keepers were also provided with free antibiotic samples or were sold antibiotics directly. A veterinarian believed that this direct marketing led to the overuse of antibiotics:


*“Here the small dairy farmer use antibiotic themselves. They use penicillin and strepto-penicillin randomly without taking any suggestions (…) I can’t say how much they are using. We forbid them to use antibiotic unnecessarily. But the medicine companies goes there and making them understand if cow is in cough and cold, give this antibiotic course, give this with that etc; they make them understand. If they come to us, I say if it is not sick no need to give it (antibiotic). But most of the time, if any sort of cough-cold seen they use penicillin, strepto-penicillin, amoxicillin, ampicillin”.*
Veterinarian 1, Site 1

Furthermore, it was the veterinarians’ opinion that livestock keepers would preferentially follow the advice of pharmaceutical representatives and self-treat rather than seek the advice of animal healthcare providers. 

In Site 2, veterinary pharmaceutical representatives were reported to additionally target providers of animal healthcare, including private veterinarians, para-vets, and public-private VPPs. Their products were retailed by drug shops in the nearby town which were used as a source of veterinary drugs by para-vets and public-private VPPs. This was explained to be the reason for the greater marketing of their products compared to Site 1. Pharmaceutical representatives reportedly maintained regular contact with animal healthcare providers through in-person meetings or by phone to promote their products:


*“They sometime come and tell that this new medicine works better in this condition. And I used that in field condition, if it works, I use it afterward (…) I am having phone number of them (like [name redacted] from [pharmaceutical company name]). Sometimes I call him, or he calls me. Whenever I am short of medicine, if I call him, he says that it could be available in this shop of [nearby town name redacted]”.*
Para-vet 2, Site 2

Para-vets and public-private VPPs in Site 2 were also provided with free product samples by representatives and, furthermore, could access discounted prices for being frequent customers of the retailing drug shops. 

### 2.3. Deterrents to Using Antibiotics

#### Antibiotics Have Side Effects

Several public-private VPPs and para-vets in both sites described experiences of side effects when treating animals using antibiotics, including abortion, irritation, and shock. Other concerns included inducement of productivity losses.


*“Once I used sulfa [sulfonamides] drugs for diarrhoea but there, abortion happened. So, I stopped using sulfa drug in diarrhoea cases”.*
Para-vet 2, Site 2

However, there was a perception that antibiotics that were more easily available were safer than other antibiotics.


*“I do not apply the potent [high power] antibiotics. I generally use easily available, safe antibiotics. In serious problem where it needs potent antibiotics, I refer to veterinary doctor”.*
Para-vet 1, Site 1

### 2.4. Crossover-Use of Antibiotics in Livestock

Use of human antibiotic formulations for therapeutic purposes in livestock systems was widespread in both study sites (Table 4). 

While it was not an objective of this study to quantify the extent of crossover use of antibiotics in livestock, some livestock keepers said they occasionally used human medicines for their livestock. Others were emphatic that they did not. The latter claim would be difficult to substantiate since livestock keepers had limited understanding of medicines and antibiotics. They were generally unaware that some drugs provided for livestock treatment might not be licensed for animal use and had implicit faith in their health providers’ ability to provide the right drugs: 


*“No, they don’t give human medication. An animal doctor would give animal medication”.*
LK5, Site 1

In contrast, many key informants, public-private VPPs, and para-vets implied that use of human antibiotics in animals was commonly practiced by them across both sites. During our field visits, we observed human antibiotic formulations stocked in the clinics of public-private VPPs in both sites. A public veterinarian said that veterinarians also sometimes prescribed human antibiotics. IPs also used human antibiotics in both sites when they were consulted for livestock healthcare. 

Human drug shops in both sites were sources of human antibiotics used in livestock via several routes. Livestock keepers could provide a veterinary prescription (either formal or informal) and the drug shop would provide a human medicine instead, or livestock keepers came to them directly to purchase medicines over the counter. In Site 1, public-private VPPs obtained human antibiotics for livestock from human drugs shops or occasionally prescribed human antibiotics for keepers to purchase themselves:


*“If it’s needed, I get it from the market [from human drugs shops]. We might prescribe it to the patient, and they get it. Sometimes we, ourselves, get it from the store”.*
Public-private VPP 1, Site 1

### 2.5. Drivers of Crossover-Use of Antibiotics in Livestock

#### 2.5.1. Lack of Access to Veterinary Drugs 

In both study sites, the closest drug shops selling veterinary medicines were in nearby towns (approximately 13 Km from Site 1, and approximately 3 km from Site 2). In Site 2, an NGO also kept a stock of veterinary medicines which were sold to livestock keepers and local para-vets. The paucity of veterinary drug shops and lack of access to veterinary drugs was a major driver for crossover use of human formulations in livestock. It was much easier for livestock keepers and public-private VPPs to obtain human antibiotics from human drugs shops than travel frequently to the more distant veterinary-specific drug shops. Both human and veterinary drug shop owners corroborated these practices: 


*“There’s no veterinary medicine shop in this area. The nearest [veterinary] shop is 6 km away”.*
Human drug shop 1, Site 1

For the public-private VPPs in Site 1, depletion of their stock of veterinary antibiotics led them to seek human antibiotics for use in livestock in order to continue serving clients:


*“In case my stock for the day is over, or I treated more patients than usual, it’d take a long time for me to go to [town name redacted], or to get it delivered from Kolkata. So, I will get the human medication from the store and get the work done”.*
Public-private VPP 2, Site 1

Depletion of veterinarian’s and VPPs’ stock of antibiotics was also a common reason for prescribing antibiotics. Provision of prescriptions resulted in livestock keepers purchasing human antibiotics from the more conveniently located human drug shops, as described in Section 2.4. The public VPP in Site 1 was reported to provide a prescription when he was unwilling to dispense antibiotics directly to a livestock keeper when treating small numbers of animals.


*“They will ask, “how many animals do you have”. If we say 10–12, they say, “we don’t have it”. They say it to our faces. If you have 50–100 animals they would, then, give the medicines. For less than that they won’t open a file of medicines”.*
LK1, Site 1

#### 2.5.2. Insufficient Veterinary Healthcare Capacity

Despite the multitude and diversity of animal healthcare providers, there were times when demand exceeded supply, resulting in livestock keepers reaching out to their nearest human sector IPs or human drug shops for emergency interventions. The public VPP in Site 1, and public veterinarians in Site 2, were available only during working hours (10 a.m.–5 p.m.) and not available on public holidays or on Sundays. Furthermore, public-private VPPs across both sites reported high caseloads. Several livestock keepers in site 1 described how they would attempt to contact either the public VPP or public-private VPPs via telephone. However, if there was no answer, they would then approach an alternative human health provider. Some IPs and human drug shop owners described only providing human antibiotics for livestock as a last resort when there was an emergency or when livestock healthcare providers were unavailable, citing a fear of causing harm to livestock, and damage to their reputation and business. 


*“Suppose he [public-private VPP] is not there, and a person came to me and asked, “Doctor my goat is having loose motion, what can we do?” If I see the condition of the goat is really bad and it might die without a treatment, I may ask him to have a human medicine of a low dose (…) but if your animal dies then I may not be responsible”.*
IP5, Site 1

Additionally, when animal healthcare providers across both sites were unavailable, some livestock keepers resorted to using spare human drugs kept in the household to treat their sick animals. These drugs were provided to them initially for the treatment of conditions suffered by members of the household. 


*“Yes, sometimes when they have fever, we give the cows our medicines… Medicines for fever, or wounds. We don’t remember exactly. The power of medicines used in humans and animals are different. We [human medicine] have less power, the medicines used in cows have high power. If we use one medicine [dose] in humans, we will use two [doses] in the cows. That we see works. When we see that doctor isn’t coming, we do it…”*
LK1, Site 1

#### 2.5.3. If It Works in People, Why Not Animals?

The use of human household drugs in livestock was influenced by the perception amongst livestock keepers in both sites that if the drugs were effective in humans, then they would likely be effective in their livestock:


*“Sometimes if we can’t buy medicines, and the goat has diarrhoea, we would use the diarrhoea medicine from the house for them to get well soon. It would probably work… We just think that if it works for humans, it might work in the cows for the same problem”.*
LK7, Site 1

During our field research, a few livestock keepers showed us some human antibiotic formulations they had used in their livestock, including ciprofloxacin (a HPCIA) to treat diarrhoea in goats. These were perceived to be effective in both humans and animals at resolving diseases with similar presenting signs (e.g., diarrhoea). 

This perception was shared by the different types of health providers, including IPs and human drug shop owners. Amongst public-private VPPs and para-vets, the only perceived differences between human and veterinary antibiotics were in their concentration of active substances and dosing requirements. Otherwise, they were considered equally effective in treating both human and livestock diseases. 


*“As much as I know, human antibiotics are 500 mg, or 250 mg, Ampicillin for example, but in case of cows, it’s 3 g or 3000 mg (…) I know this much that a cow would need a higher dosage. Both of them are of the same Ampicillin group, but veterinary medicine is very powerful, and human medicines are just 500 mg…”*
Public-private VPP 1, Site 1

#### 2.5.4. Some Human Antibiotics Are Better

In Site 2, some veterinarians and para-vets reported using human antibiotics due to the perception that these medicines were more effective: 


*“Norfloxacin gets some better result in case of goat (…) When the norfloxacin is not available we use enrofloxacin (…) According to size and body weight (…) this type of medicine is available in the market in 200 mg and 400 mg preparation, in the case of adult goat we are using 400 mg like a human dose”.*
Veterinarian 5, Site 2

One para-vet believed this to be due to human antibiotics being of higher quality in comparison to their veterinary counterparts:


*“I have seen some human medicine works very much in animal (…) I have seen in mastitis my medicine is not working but Clavam [amoxicillin clavulanate] works (…) Quality of human antibiotics is better”.*
Para-vet 3, Site 2

One public veterinarian additionally stated that a reason for getting better results using human formulations was that they had not yet been used in animals.

#### 2.5.5. Higher Dosing Accuracy of Human Antibiotics in Small Livestock Species and Other Animals in the Household

Several public-private VPPs in Site 1 explained that the ease of administering a correct dose of antibiotics for small ruminants, poultry, and pets was a reason for opting for human antibiotics instead of their veterinary counterparts. They also perceived the veterinary antibiotics available to them to be of too high a concentration to use in those animal species.


*“In cases of small animals, like dogs, goats etc. the veterinary antibiotics are of high power. For example, they are 3 g. I would need 1 g. So, I can’t use them, so I need to use human antibiotics”.*
Public-private VPP 1, Site 1

### 2.6. Deterrents to Using Human Antibiotics in Livestock

#### Perceptions of Livestock Keepers That Human Antibiotics Are Unsuitable for Use in Livestock

While many livestock keepers were content to use human medicines in livestock, and were not concerned of any potential side effects, others were averse to this practice since they saw humans and livestock to be fundamentally different: 


*“We shouldn’t give human medicines to animals (…) It won’t suit them. If we take their medicines, it won’t suit us. If they take our medicines, it won’t suit them (…) We have a different stomach (…) that’s why it won’t suit”.*
LK2, Site 1

Just as human drug shop keepers were cautious regarding use of human medicines in animals, some livestock keepers were also reluctant to give human drugs to their livestock for fear of causing harm. 


*“The difference is that humans can talk, we can express which problems we have. But the animals can’t talk. We can feed the medicines, but they won’t be able to tell us if the medicine is causing a problem. We are scared that human medicines might harm animals”.*
LK4, Site 1

Others expressed the opinion that humans and livestock suffered from different diseases which required different treatments, just as the healthcare providers sought for humans and animals were different.


*“To get medicines for humans we need to go to a different doctor and for the animals to a different one. (…) A man may have pox, similarly a cow or a chicken may have pox as well. Since the species are different, therefore medicines must be different as well. When a chicken has diarrhoea, the medicine is different than when a man has diarrhea”.*
LK10, Site 1

Several livestock keepers expressed concern that human formulations had a lower ‘power’ (strength) than veterinary formulations, and so would be unlikely to successfully treat livestock diseases. 


*“No, we would never use human medicines on animals. The cow-tablet is this big, and the tablet for human is just this small. Of course, there’s a difference (…) The animal medicines have higher power”.*
LK3, Site 1

### 2.7. Crossover Use of Veterinary Antibiotics in People

No evidence was found to suggest that veterinary antibiotics were being used to treat people in either site. No antibiotic provider reported supplying veterinary antibiotics nor did livestock keepers report using any surplus veterinary antibiotics for human treatment. 

### 2.8. Deterrents to Using Veterinary Antibiotics in People

#### 2.8.1. Veterinary Antibiotics Have a Higher Power and Are Harmful to Humans

Across both sites, many livestock keepers themselves expressed an awareness of differences between human and veterinary medicines. They explained that veterinary medicines were of higher “power”, different composition, or generally unsuitable for human consumption. These higher ‘power’ medicines were deemed harmful to humans. Veterinary medicines were reportedly stored outside of living spaces and kept away from children in the household, with one livestock keeper believing that use of veterinary medicines in humans could have fatal consequences:


*“(…) the medicine for the chickens are to be given to the chickens. Those who want to commit suicide, they would take such medicines”.*
LK10, Site 1

#### 2.8.2. They Are ‘Not for Human Use’

Veterinary antibiotics that were stocked by the different providers we interviewed had clear warnings against their use in humans. These warnings may have had an impact on the providers and prevented them from using veterinary antibiotics in humans: 


*“In veterinary antibiotic, it is written there that ‘not for human use’. There is no question of giving [to people]”.*
Para-vet 2, Site 2

However, human antibiotics lacked any statements on their packaging warning against their veterinary usage. 

## 3. Discussion

Our study provides new evidence for ABU and its drivers in smallholder production systems in India that have received scarce attention in ABR literature, despite these systems accounting for 85% of its livestock production [9]. Humans and animals shared habitats, antibiotics, and certain health providers. Antibiotics were used mostly therapeutically in small holder settings but were used inappropriately by both livestock keepers and antibiotic providers, including veterinarians. Of greatest concern was the routine usage of (HP)CIAs, chloramphenicol usage in poultry, and crossover-use of human antibiotics in animals. Our findings have important One Health implications for ABR and antibiotic stewardship in rural community settings. 

Consistent with our findings, previous studies among smallholder dairy farmers in numerous East Indian states have also indicated that antibiotics are predominantly used for therapeutic purposes in livestock [13,15]. In contrast, in commercial settings in India, and worldwide, they are often used for disease prevention and growth promotion [7,8]. Compared to this unnecessary usage, therapeutic usage in smallholder livestock, as identified in our study and others, would appear to be more rational. However, therapeutic ABU can still be inappropriate, and our study details the different practices that can be considered as such including the provision of short courses and experimental treatments amongst others. Similar inappropriate practices have also been observed in human health in our setting in a linked study [17]. One implication would be that humans and animals within smallholder settings are particularly at risk of ABR developing can transmit across species and into the environment [2,5]. 

Different cadres of privately acting VPPs in both sites were critical sources of animal healthcare and antibiotics to livestock keepers, despite not being legally authorised to provide antibiotics. They were used because public veterinary services were either inaccessible or unavailable, which aligns well with the findings of Swai et al. (2021) that one of the crucial factors effecting healthcare decision making is access to resources [32]. The current shortfall in field veterinarians in India is 35,500 less than the 75,000 required [33]. This shortage has been described as the major problem leading to the inadequate use of antibiotics by dairy farmers in several locations in India and is exacerbated in rural areas [16]. Public-private VPPs and para-vets filled this gap in animal health services in our study, as previously noted in comparable settings in India [13,15,16]. However, our findings suggest that simply increasing veterinary capacity may be insufficient towards reducing inappropriate ABU, as veterinarians were equally found to engage in inappropriate practices as less qualified VPPs. Therefore, any efforts to improve antibiotic stewardship in these settings must target the knowledge and practices of both veterinarians and VPPs.

There is a glaring lack of guidelines for veterinary ABU for therapeutic purposes in India, even for veterinarians. Various state agencies have sporadically issued recommendations for reducing non-therapeutic ABU for growth promotion and prophylaxis [34,35,36]. However, these recommendations are aimed at commercial food animal production, and do not extend to the treatment of individual animals or small-scale production systems. If disseminated and made available to a wider range of animal healthcare and antibiotic providers, such as formal and informal animal healthcare providers, IPs and drug shop owners, these could serve to raise awareness for judicious ABU. Similar antibiotic stewardship interventions have been conducted in human health settings with positive results [37,38]. Durrance–Bagal et al. [39] and Swai et al. (2010) [40] argue that a One Health approach involving human and animal healthcare providers as well as community members is necessary when developing interventions to mitigate the risk of zoonotic diseases. Considering the multitude of actors involved in the delivery of animal healthcare to livestock in these settings, and the impact of ABR across sectors, a similar, multi-stakeholder approach to the design of guidelines will likely be necessary. This should include those not authorised to prescribe antibiotics but who, nevertheless, deliver critical animal health services, as we found in our study. This is contrary to India’s NAP-AMR which promotes enforcement of regulations prohibiting unauthorised antibiotic provision [41]. However, this legislation would result in the imposition of further barriers for livestock keepers’ access to already limited animal health services. Ongoing stakeholder consultations have been conducted as part of our wider project. Key outcomes have been the recommendation to bring together local governmental and non-governmental stakeholders for guideline development, feasibility testing and implementation. Epidemiological assessments on antibiotic sensitivity patterns and the views of local professional and paraprofessional providers must also inform and guide the formulation of guidelines, for these to be effective.

An additional strategy could involve implementing the OIE Performance of Veterinary Services (PVS) report for India which recommends greater supervision of VPPs by veterinarians to ensure their responsible ABU [9]. However, the shortage of veterinarians in our setting challenges their effective supervision. Alternative strategies, such as the use of digital communication through social media or mobile applications, could be considered to circumvent this issue. Mobile health applications have increased adherence to antibiotic guidelines in human hospital settings and led to reductions in ABU [42]. Such innovative options could be adapted for One Health antibiotic stewardship in our settings. 

The competitive market for antibiotic provision exacerbated inappropriate ABU and has previously been identified as one of its drivers in India [13,43]. The public-private VPPs and para-vets lacked incentives to act as antibiotic stewards in a competitive private health market. The economic incentives to provide antibiotics could represent one of the most challenging barriers towards implementing a successful antibiotic stewardship intervention in our settings. Marketing of antibiotics by veterinary pharmaceutical representatives targeting livestock keepers, veterinarians, and VPPs observed in our study, that has also been reported in smallholder dairy systems in India, adds another layer of complexity to the situation [13]. The language of the current regulation prohibiting the advertisement of medicines is open to interpretation and we would echo previous calls to review and refine the current regulations to help mitigate inappropriate ABU [13,44].

Of great concern was that most of the antibiotics identified and reportedly used in our study were (HP)CIAs. The WHO has deemed that (HP)CIA use be restricted both in human and veterinary medicine to protect their therapeutic efficacy [45]. There are concerns that their overuse in animals may result in resistance spreading to bacteria of public health significance [27,46]. In our study, (HP)CIA usage in livestock was ubiquitous among all types of antibiotic providers. Veterinary diagnostic laboratories were absent or inaccessible to antibiotic providers, a finding echoed in other studies [47]. Selection of appropriate antibiotics based on antibiotic susceptibility testing was therefore impossible. In the face of diagnostic uncertainty, strategies to optimise empirical antibiotic therapy could simply begin with a reduction in the usage of broad spectrum (HP)CIAs in animals.

Chloramphenicol usage was also identified in poultry and is a public health concern. Its residues in food are potentially carcinogenic and can cause non-regenerative aplastic anaemia in humans [48,49]. Its usage is banned in food-producing animals in the EU [50] and in many other countries worldwide. In India, it is banned for usage in aquaculture [51] and at any stage of egg powder production intended for export [52,53]. However, no regulations exist for chloramphenicol usage in food-producing animals intended for the national market [53]. In our study, chloramphenicol usage in poultry could represent a food safety risk to consumers due to the potential persistence of their residues in meat. Its usage in food animal production systems warrants further investigation in India. 

The crossover-use of human antibiotic formulations was identified in livestock, including HP(CIAs). To our knowledge, this is the first study to directly investigate and report this phenomenon in India, although it has been reported in other countries including Guatemala, Nigeria, Uganda, and Cambodia. Furthermore, there is limited evidence of this practice in the current literature. Nevertheless, it has been reported that smallholder farmers in Guatemala used human antibiotics in poultry due to poor access to veterinary services and drugs, and a perception that human antibiotics are as effective as veterinary antibiotics at treating poultry diseases [24]. In Uganda, Nigeria, and Cambodia, human-intended antibiotic products were purchased from human drug shops for animal use [22,23,25]. Snively-Martinez (2019) argues that this practice could feasibly result in increased ABR infections in people, given previous evidence of high levels of bacteria resistant to human antibiotics in smallholder poultry production [24,54,55]. Other than potentially contributing to ABR, the crossover-use of human antibiotics in livestock may risk inducing harm in animals as human and veterinary drugs are often formulated differently to accommodate the differences in physiology, and thus pharmacokinetics of a given drug, between species. The actual efficacy and safety of human formulations in animals is unknown [28]. Only limited studies have explored the adverse effects of human formulation use in animals including chloramphenicol in piglets, ceftriaxone in horses, and on food animal productivity (e.g., prolonged sulphonamide use in poultry causes thinning of eggshells and a loss of profit for farmers) [29]. Further research is required to understand the potentially harmful effects of crossover-use and its causal pathways.

To mitigate crossover-use in this setting, the drivers identified can be addressed through practical solutions. Animal healthcare providers require improved access to veterinary antibiotics and livestock keepers need better access to veterinary healthcare and drugs. A broader variety of packaging of veterinary drugs should be available to livestock keepers and animal healthcare providers to enable treatment of large, small, or few, animals with the correct drugs. Public-private partnerships with pharmaceutical companies could assist the distribution of veterinary pharmaceuticals and improve the convenience of packaging. Labelling of veterinary antibiotics was found to deter their human use in our study. Equivalent labelling warning against the usage in animals of human formulations could raise awareness in antibiotic users.

In the shorter-term, interventions that include educating livestock keepers and antibiotic providers about responsible ABU and crossover-use are needed [40]. Previous studies suggest this as a strategy to reduce crossover-use of human chloramphenicol [25]. Responsible ABU should be included in the initial training of public and public-private VPPs, and para-vets. In cases where crossover-use is required, they could be encouraged to limit themselves to antibiotics in the ‘Access’ group, as defined by the WHO [56]. 

Promoting good animal husbandry, biosecurity practices and disease prevention and control in animal populations could be another effective strategy in our study setting towards reducing ABU [57]. Farmer Field Schools (FFS), as recommended by FAO, are model farms that provide context-specific curricula for local needs [58]. This initiative also trains local farmers to become facilitators, thereby encouraging a more tailored and sustainable system of smallholder agricultural education [58]. In India, FFS have been successfully implemented in crop farming [59] and existing infrastructure could be adapted for promoting sustainable animal production and responsible ABU in our study setting. 

Other alternatives in India include the Krishi Vigyan Kendras (KVKs). They serve as knowledge networks to develop local agricultural economies through the assessment and provision of location-specific agricultural technologies and have been considered responsible for improving agricultural knowledge amongst farmers in many of India’s districts [60,61]. 

Vaccination uptake amongst livestock keepers is unknown in our specific sites and was beyond the scope of our study. However, livestock vaccination is reportedly lower (20%) in West Bengal than other eastern Indian states [15]. Considering that vaccination is regarded as an effective strategy to reduce the burden of infectious diseases and reduce the overreliance on antibiotics in livestock [62], further research could determine whether its improvement could optimise ABU in our study sites. 

There were some limitations to this study. We did not quantify the frequency and volumes of antibiotics that were used in animals, nor of crossover-use. Future research is required in these areas to understand the magnitude of the challenge. Our study reflects the situation in two rural contexts in West Bengal, India, and may not be generalisable to other regions with different socioeconomic contexts and production systems. Livestock keepers’ understanding of antibiotics made investigation of their usage challenging at times and our assumptions and interpretations of their responses might not be fully accurate. Nevertheless, the range of respondents allowed us to triangulate individual responses to ensure validity of findings. Interviewers often asked participants to describe past events or behaviours and there is a risk that the findings are subject to recall bias. Furthermore, certain questions may have been sensitive as they probe practices that were not legal, so it is possible that their responses were prone to social desirability bias. However, rapport was developed before commencing interviews, participants were interviewed in familiar settings, and open-ended questions were asked to reduce this bias. 

## 4. Materials and Methods

This study forms part of a wider project “A multi-stakeholder approach towards operationalising antibiotic stewardship in India’s pluralistic rural health system” which aims to co-design and pilot an intervention that generates an antibiotic stewardship programme to promote responsible ABU in West Bengal, India. Our sub-study aims to contribute by interviewing diverse stakeholders using a ‘One Health’ approach to explore the drivers of ABU in animals of smallholder settings, and crossovers between animal and human health. 

### 4.1. Study Setting

This study was conducted in the state of West Bengal (Figure 1). West Bengal is the fourth most populous state in India, with most of the population living in rural settings (68.1%) comparable to the country’s average (68.8%) [63]. West Bengal has a large livestock sector, ranking third among the states in meat and egg production and 11th in the production of milk. Livestock represents an important industry as it constitutes 3.89% of the state’s domestic product, and 20.3% of agricultural production [64]. There is no specific livestock census data at village level in West Bengal; however, the latest livestock census in 2012 reported that 14,481,194 (85.4%) out of a total of 16,953,359 rural households (around 85%) in the state-owned livestock [65]. The predominant livestock species kept are cattle (n = 6,321,601), poultry (n = 5,534,883), and goats (n = 3,807,719) [65].

Villages were selected from two different Gram Panchayats (GP: village administrative unit consisting of a cluster of villages) in the South 24 Parganas district of West Bengal. Hereafter, the two GPs are referred to as Site 1 and Site 2. Based on estimates obtained through an earlier study, villages were selected in each of the two sites that had more than 50% of households owning livestock. Both sites also had active informal providers (IPs), a cadre of community-based human healthcare providers with informal paramedical training [17]. Selecting two separate sites enabled comparisons to be made between livestock healthcare, livestock systems, and the provision and usage of antibiotics between the two sites. The number of government veterinarians at district-level is not publicly available. However, personal communication with key contacts in the district indicated that there were 102 government sanctioned veterinary posts in South 24 Parganas, seven of which were currently vacant. With 95 active government veterinarians in the district, there was one government veterinarian per 85,915 population head based on the population of the district [63].

In Site 1, villages were closely clustered, and two villages were selected that met the selection criteria. Site 1 is located within Diamond Harbour II block 60 km by road from Kolkata and bordered by the river Hooghly. A block is a rural area subdivision of a district for the purpose of rural development and covers several GPs. The populations of the villages studied in Site 1 were 7527 and 1379.

Site 2 was larger, almost twice the area of Site 1, and villages were more scattered in comparison. Four villages were selected in Site 2. Site 2 is located within Gosaba block, 95 km by road and boat from Kolkata, across the river Matla at the edge of the Sundarbans national park. In Site 2 the village populations were 3620, 3946, 3673, and 3467 [63]. 

In South 24 Parganas, agriculture, forestry, and fishing (48%); wholesale and retail trade (10%); and construction (6%) were the largest industries by number of main workers [63]. Information on the level of education within specific villages is not publicly available but adult literacy rates for South 24 Parganas is 87.3% [63]. 

### 4.2. Selection of Participants

Categories of interviewees included local key informants, livestock keepers, and antibiotic providers. Key informants were introduced to us by our local collaborators and university partners and were selected for their anticipated high level of knowledge of the study communities, including details of livestock systems, related health practices, and antibiotic knowledge. These interviews helped to set-up a preliminary list of animal health providers and livestock keepers in the villages, with further antibiotic providers and livestock keepers being identified using a snowballing approach (i.e., participant selection was informed by previous interviews). Livestock keepers were inhabitants of the villages in the study sites who raised animals, usually for personal consumption. Antibiotic providers were defined as healthcare providers who serviced the study sites and supplied and/or prescribed antibiotics to livestock keepers for use in their livestock or administered medicine directly to livestock. 

### 4.3. Data Collection

Tailored semi-structured interview guides were designed for key informants, antibiotic providers, and livestock keepers with the key topics for each interviewee category presented in Table 5. 

Interviews were conducted across the study sites from June 2019–January 2020. Piloting of the interview guides was conducted as part of initial interviews with livestock keepers and antibiotic providers (two from each category). Adjustments were made to capture richer information and enable better cohesion of subsequent interviews.

The majority of interviews (64) were conducted by three researchers trained in qualitative research, each accompanied by a local research assistant who translated and made notes. Furthermore, interviews were conducted with three IPs in Site 2 by our local collaborators in India. Interviews typically lasted between 30 min to one hour and were conducted in Bangla. When consent was provided by participants, interviews were voice recorded and contextual photographs taken.

To elucidate information regarding ABU, providers were asked which antibiotics they dispensed and, where possible, to show their stock of both human and veterinary medicines. Livestock keepers were asked to show any medicines (or empty medicine packaging) used in their livestock that were still on the premises for triangulation purposes of ABU data collected through the interviews. For the purpose of this paper, crossover use has been defined as the use of licensed antibiotic formulations for humans in animals and vice-versa. In some countries, this may also be known as “off-label” or “extra label” use.

Photographs were taken across both sites to support interview findings and to enrich our data. These included photos of smallholdings, shared household spaces with livestock, and antibiotic and non-antibiotic medicines (and their packaging) found on the premises of livestock keepers and reported to have been used in animals. Stocks of antibiotics maintained by antibiotic providers were also photographed. Photographs can be found as Appendix A.

### 4.4. Data Analysis

Interview audio recordings were transcribed and translated into English. Thematic analysis was conducted on all transcribed interviews and photographs following the process developed by Braun and Clarke [66]. This process involves six distinct phases: (1) familiarisation with the data; (2) generating initial codes; (3) searching for themes; (4) reviewing themes; (5) defining and naming themes; and finally (6) contextualising and discussing the themes. 

The coding was undertaken using NVivo v12 (QRS International). A combination of deductive and inductive approaches was used to develop the codes. The coding process was validated by three senior researchers, experienced in qualitative data analysis. An iterative approach was utilised throughout the analysis stage until a final set of codes and themes were agreed upon.

## 5. Conclusions

Our study describes the way in which antibiotics are used in livestock in two rural sites in India, highlighting widespread crossover-use between human and livestock sectors which is a new finding in the existing India literature. Although antibiotics were used mainly for therapeutic purposes, and prophylactically only in small-scale commercial poultry farms, many of the antibiotic practices identified were inappropriate. These practices were common amongst veterinarians and VPPs, and were influenced by multiple social, economic and policy related drivers. Antibiotic stewardship cannot be achieved in this setting by merely increasing the number of veterinarians; there is an urgent need to harness all available providers by designing proper evidence-based ABU guidelines for all levels and allowing VPPs access to some essential veterinary antibiotic formulations. Informal providers of human health and drug retailers, along with communities, need to be made aware of the issues associated with crossover-use. The existing mentorship links between the formal and informal must be harnessed for improved overall practices and the pharmaceutical industry’s support be sought for manufacturing and making available more economically packaged veterinary antibiotics, and appropriately labelled human antibiotics. All this calls for a multi-sectorial—community, provider, health system, and pharmaceutical level—One Health intervention to operationalise antibiotic stewardship in these settings in the near future. Long term strategies will need to focus on the production of appropriately trained VPPs and veterinarians in larger numbers who can meet the animal health needs of the rural poor adequately, and expansion of veterinary diagnostic testing and surveillance capacity. 

## Figures and Tables

**Figure 1 antibiotics-10-01433-f001:**
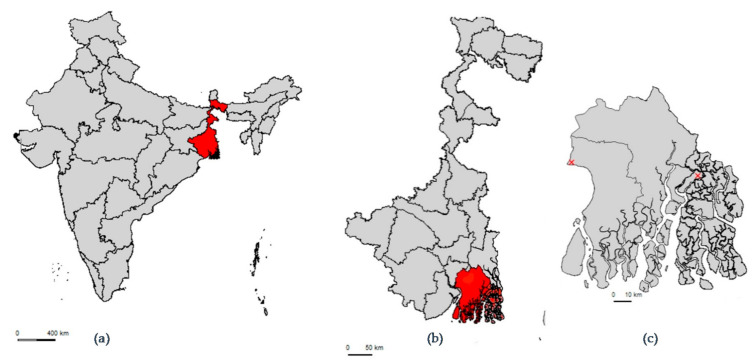
(**a**) Map of India highlighting West Bengal. (**b**) Map of West Bengal highlighting South 24 Parganas. (**c**) Map of South 24 Parganas marking Site 1 (West) and Site 2 (East).

**Table 1 antibiotics-10-01433-t001:** Summary of the participants interviewed (n = 67).

Type of Interviewee	Site 1	Site 2
Key informants (n = 9)	Private veterinarian (n = 1) ^1^ High school teacher (n = 2) Ex-village chief (n = 1) Homeopath (n = 1)	Veterinary officer (n = 3) ^1^ Private veterinarian (n = 1) ^1^
Antibiotic providers (n = 26)	Private veterinarian (n = 1) ^1^ Animal Development Volunteer (n = 1) Pranibandhu (n = 1) Para-vet (n = 1) Veterinary drug shop (n = 2) Human drug shop (n = 1) Informal provider of human health (n = 5) Homeopath (n = 1)	Veterinary officer (n = 3) ^1^ Private veterinarian (n = 1) ^1^ Para-vet (n = 2) Pranibandhu (n = 1) Pranimitra (n = 1) Poultry Shop (n = 1) Human drug shop (n = 1) Informal provider of human health (n = 3)
Livestock keepers (n = 37)	n = 23	n = 14

^1^ Interview participants who served as both key informants and antibiotic providers.

**Table 2 antibiotics-10-01433-t002:** Typology and description of the antibiotic providers in the study sites.

Classification of Antibiotic Provider	Description of Classification
Veterinary Officer	A government employee who has received a university degree in veterinary medicine
Private veterinarian	A self-employed worker who has received a university degree in veterinary medicine
Public veterinary paraprofessional (Public VPP) ^1^	A government employee who has received formal, longer term (≥six months) training from the government or recognised academic training institution in livestock services and primary veterinary care
Public-Private veterinary paraprofessional (Public-Private VPP)	A livestock healthcare provider who has received short term (≤six months) formal training from the government in livestock services (predominantly artificial insemination) and works in a dual public/private capacity
Para-vet	A self-employed animal health worker informally trained in primary veterinary care
Homeopath	A self-employed health worker trained in human homeopathic medicine
Informal provider of human health (IP)	A self-employed health worker who does not hold a medical degree but is informally trained in the practice of human medicine
Human drug shop ^2^	A shop that sells allopathic medicines that are manufactured with the intention of human consumption
Veterinary drug shop	A shop that sells allopathic medicines that are manufactured with the intention of animal consumption
Poultry Shop	A shop that sells poultry-specific agro-veterinary supplies including allopathic veterinary medicines

^1^ The LDA in Site 1 (a public VPP) declined to be interviewed but was identified as an antibiotic provider to livestock keepers of Site 1. Findings presented relating to his practices were identified indirectly through interviews with key informants and livestock keepers. ^2^ Some human drug shops supplying livestock keepers in Site 2 contained veterinary sections.

**Table 3 antibiotics-10-01433-t003:** Major Differences between Site 1 and Site 2.

	Site 1	Site 2
Geographical	- 60 km from Kolkata (the state capital), accessible in 2–2.5 h by road - Closer to urban and peri-urban areas - Smaller area - Villages more closely clustered	- 90 km from Kolkata, accessible in 4 h by road and a 20 min ferry ride - Further from urban and peri-urban areas - Larger area - Greater dispersion of villages
Livestock Production Systems	- Predominantly small-holder production systems catering for household use and supplementary income - Smaller number of small-scale poultry production systems	- A greater number of small-scale commercial poultry production systems, catering to the local hospitality industry and Sunderban tourist trade
Veterinary Services	- No public veterinarian nearby - One private veterinarian, 13 km away - One public VPP - Multiple public-private and private VPPs	- Three public veterinarians, two active and one with a largely administrative role - One private veterinarian operating from a local NGO within 3 km - Multiple public-private and private VPPs
Drug Shops	- Two veterinary drug shops, both 13 km away - Multiple human drug shops, none stocked veterinary medicines	- One veterinary drug shop (poultry shop) 7 km away - Multiple human drug shops, some of which stocked veterinary medicines

**Table 4 antibiotics-10-01433-t004:** Veterinary and human antibiotic formulations identified and used in livestock.

Antibiotic Class	Antibiotic Formulation ^1^	Livestock Species
Aminoglycoside	Gentamicin *^,C^	Cattle, Poultry, (Dogs)
Cephalosporin	Ceftiofur ^H^, Ceftriaxone ^H^, Ceftriaxone-Sulbactam ^H^	Cattle
Fluroquinolone	Ciprofloxacin *^,H^, Enrofloxacin ^H^, Marbofloxacin ^H^, Norfloxacin *^,H^, Ofloxacin ^H^	Goats, Poultry, Sheep
Macrolide	Azithromycin *^,H^, Tylosin ^H^	Poultry, (Dogs)
Nitroimidazole	Metronidazole *	Cattle, Goats, Poultry
Penicillin	Amoxicillin ^C^, Ampicillin ^C^, Amoxicillin-Clavulanate *^,C^, Ampicillin-Cloxacillin ^C^, Penicillin *	General
Phenicol	Chloramphenicol *	Poultry
Sulfonamide	Sulfadiazine *	Poultry
Tetracycline	Oxytetracycline *, Tetracycline *,	General
Trimethoprim	Trimethoprim	- ^2^
Combination	Neomycin sulphate-Bacitracin ^C^, Ofloxacin-Ornidazole *^,H^, Trimethoprim-Sulfamethoxazole *	General

^1^ Antibiotics marked with a * indicates human formulations found to be used in livestock, ^C^ indicates formulations that are CIAs, and ^H^ indicates formulations that are HPCIAs. ^2^ Cells left blank indicate that data regarding the species this formulation was used in were not obtained.

**Table 5 antibiotics-10-01433-t005:** Key interview guide topics for the different interview stakeholders.

Type of Interviewee	Interview Guide Key Topics
Key informants	- Identification and characterisation of the livestock systems - Identification and characterisation of the different healthcare providers for livestock and humans - Drivers of ABU and provision in livestock - Investigation of potential overlaps of human and livestock ABU at the household or farm level
Livestock keepers	- Identification and characterisation of the livestock system - Investigation of ABU: sources and drivers of usage - Investigation of potential overlaps of human and animal ABU at household and farm level
Antibiotic providers	- Characterisation of antibiotic providers and their role within the community - Investigation of antibiotic dispensing and drivers of usage - Investigation of potential overlaps of human and animal ABU at provider level - Investigation of antibiotic provider level of training and affiliations

## Data Availability

The data used and/or analysed during the current study are made available as Appendix A.

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
