# Peer review of "“If It Works in People, Why Not Animals?”: A Qualitative Investigation of Antibiotic Use in Smallholder Livestock Settings in Rural West Bengal, India"

_antibiotics, 2021, doi:10.3390/antibiotics10121433_

Round 1

Reviewer 1 Report

The work by  Arnold and colleagues describes a very critical situation with a high impact on local (and virtually global) health. Although the scientific intent is very noble, the lack of elaboration with epidemiological parameters (risk characterization, analysis of variables, use of multivariate models) does not allow any conclusion to be drawn that is supported by statistical interpretation, but only a description of a situation (the Indian one) that is widely known for its lack of epidemiological-health surveillance.

Author Response

Point 1.  

The work by Arnold and colleagues describes a very critical situation with a high impact on local (and virtually global) health. Although the scientific intent is very noble, the lack of elaboration with epidemiological parameters (risk characterization, analysis of variables, use of multivariate models) does not allow any conclusion to be drawn that is supported by statistical interpretation, but only a description of a situation (the Indian one) that is widely known for its lack of epidemiological-health surveillance. 

Response to Point 1.

We would like to thank the reviewer for the time taken to review our manuscript and for the feedback provided. However, we would like to clarify that our study was a qualitative study by design and for this purpose we adopted social sciences methods.  We did not collect quantitative data in order to conduct the analyses described in the comments provided as this was not part of the scope of the study. Our study has immense relevance as very little is known about the reasons that antibiotics are used in rural smallholder settings in India, or about the actors providing them. Our study aimed to fill in this gap in knowledge and to gain a better understanding of the specific context and the behaviours and practices for antibiotic usage that can subsequently inform future epidemiological studies. We focused on the identification and characterisation of the different sources and mechanisms of antibiotic provision to smallholder farmers in our sites, and to understand the drivers of antibiotic usage in these systems. For these aims, social science and qualitative methods are considered as necessary in the study of infectious diseases and AMR as quantitative studies, since they can answer questions that quantitative studies are unable to. 

Our method of data collection was primarily through conducting semi-structured interviews that enabled an in-depth enquiry into the perspectives of the stakeholders and gave us a rich understanding of their behaviours as well as explanations for these behaviours.  Quantitative methods would not be suitable for this kind of enquiry. 

Qualitative analysis of the data was through thematic analysis as detailed by Braun and Clarke (2006), a widely used methodology for the analysis of qualitative data described in the methods section (lines 1072-1082, page 24) (see ref 1).  This methodology involves the categorisation of interview data into codes from which wider themes and sub-themes are identified. Although subjective in nature, the coding process and identification of themes was validated by multiple researchers (lines 1078-1082, page 24). The findings we present in our manuscript (drivers) therefore represent themes identified for the two study sites alongside more specific findings in our descriptive analysis. Qualitative methods do not require statistical data analysis in order to for the findings to be valid, in comparison to quantitative studies that require results to be statistically significant. It should also be noted that quantitative methods of study to understand how antibiotics are used in these settings is challenging. Livestock keepers had very little awareness of antibiotics or the names of medicines they administered to their animals and rarely kept packaging. Furthermore, providers of antibiotics were largely informal and did not keep medicine records or receipts of sales. Although quantifying the quantities and frequency of antibiotic usage was not an objective of our study, alternative study designs would be required that take into consideration the challenges these settings impose to capture these type of data. 

Qualitative research studies allow researchers to gain a more in-depth understanding of the different contexts where antibiotics are used in livestock, even within the same country, where their usage and its drivers are likely to be different. The findings can subsequently provide valuable information towards the design and implementation of interventions that are appropriate and effective in the specific context. It is worth mentioning that our study was conducted as part of a wider project, “One Health Antibiotic Stewardship in Society” (OASIS), to inform further studies in the project. These included an intervention study and a value chain analysis study

  1. Reference: Braun V, Clarke V. Using thematic analysis in psychology [Internet]. 2006 [cited 2019 Jul 11]. Available from: https://core.ac.uk/download/pdf/1347976.pdf 

Reviewer 2 Report

The article has followed the scope of the journal Antibiotics and readers of the journal will be benefited; however, the author needs to work on the presentation of the results and edit the unnecessary discussion to make it more exciting for the readers.

I would recommend the article could be published in Antibiotics, after a minor revision.

The authors need to address the below-mentioned queries.

  1. The Author could provide the geographical details (not name) of the two villages along with total population education, and source of income.
  2. The Author needs to concise the results and discussion part and focused on representing the key point in tabular form and separate them into two parts as site 1 and site 2 for each section. There was a lot of repetition of similar facts, so the author could highlight the important point for discussion as interview transcripts are provided in supplementary.
  3. Supplementary tables need to combine in one file with footnotes.
  4. All the photographs need to combine and provide captions for each one.
  5. The author could divide the interview transcripts into two files as site 1 and site 2.
  6. The author could make comments on if there is any similarity between the two sites or any major difference between the two sites. And how these factors effecting on controlling the use of antibiotics for livestock.
  7. Reference should be in the same format.
  8. The author could include the following references:

(a) Sharma, G.; Mutua, F.; Deka, R.P.; Shome, R.; Bandyopadhyay, S.; Shome, B.; Goyal Kumar, N.; Grace, D.; Dey, T.K.; Venugopal, N.; et al. A qualitative study on antibiotic use and animal health management in smallholder dairy farms of four regions of India. Infect. Ecol. Epidemiol. 202010, 1792033

(b) Durrance-Bagale A, Rudge JW, Singh NB, Belmain SR, Howard N. Drivers of zoonotic disease risk in the Indian subcontinent: A scoping review. One Health. 2021;13:100310. Published 2021 Aug 14. doi:10.1016/j.onehlt.2021.100310

(c) SwaiE. S., SchoonmanL., & DabornC. (2010). Knowledge and attitude towards zoonoses among animal health workers and livestock keepers in Arusha and Tanga, Tanzania. Tanzania Journal of Health Research12(4), 272-277. https://doi.org/10.4314/thrb.v12i4.54709

(d) António Teixeira Rodrigues, Fátima Roque, Amílcar Falcão, Adolfo Figueiras, Maria Teresa Herdeiro Understanding physician antibiotic prescribing behaviour: a systematic review of qualitative studies,

International Journal of Antimicrobial Agents, Volume 41, Issue 3,

2013, 203-212, https://doi.org/10.1016/j.ijantimicag.2012.09.003.

Author Response

Point 1.  

The Author could provide the geographical details (not name) of the two villages along with total population education, and source of income. 

Response to point 1.

We have now added this information in our methods section, including village-level population data, the level of education (adult literacy rate) and the main industries/sources of income in the areas of study. Please see:

Lines 991-1007, page 21-22

Point 2.

The Author needs to concise the results and discussion part and focused on representing the key point in tabular form and separate them into two parts as site 1 and site 2 for each section. There was a lot of repetition of similar facts, so the author could highlight the important point for discussion as interview transcripts are provided in supplementary. 

Response to point 2.

In response to this comment, we have made several changes to our results and discussion sections towards making these more concise (such as removing unnecessary details and through rephrasing of sentences). Regarding the differences between the two sites, we found that these differences were limited to a few socio-demographic characteristics, type of livestock production systems and access to the veterinary healthcare infrastructure including drug shops. We have highlighted these in Table 3 (see line 261, page 7) However, the findings about antibiotic use and its drivers were very similar across both sites. We have edited these to make them more concise. We have also edited the discussion to focus on our key findings and their implications and to reduce word count.  Our discussion is now reduced to 2085 words from the earlier 2583 words.

Point 3.

Supplementary tables need to combine in one file with footnotes. 

Response to point 3.

Supplementary tables are now under the same folder S1, each table is in a separate word document and named appropriately.

Point 4.

All the photographs need to combine and provide captions for each one. 

Response to point 4.

Supplementary photographs are now under the same folder S3 and have individually been captioned to describe what they are providing evidence for

Point 5.

The author could divide the interview transcripts into two files as site 1 and site 2. 

Response to point 5.

Interview transcripts are now collected in the same file S2. Within this, they have been separated into sites 1 and 2 and been individually captioned to specify the interviewee details

Point 6.

The author could make comments on if there is any similarity between the two sites or any major difference between the two sites. And how these factors effecting on controlling the use of antibiotics for livestock. 

Response to point 6.

We have partially addressed this comment in our response to comment 2 of Reviewer 2. We have described the broad contextual and infrastructural differences in table 3 on page 261, and also attempted to make these differences more explicit throughout the results section. However, as most of the themes identified were common across both sites, our points for how antibiotic stewardship could be ameliorated in the discussion remain largely unaffected. It must be noted that these two sites, even though quite far apart were in the same district, governed by the same administration and so the differences were unlikely to be significant enough to affect the way that antibiotics were used. What this does show is that rural India can be quite diverse and still quite similar in the way that antibiotics are used. In fact, some of our findings about animal treatment by paravets and the drivers of inappropriate use are similar to those reported in the state of Haryana, which is quite far from West Bengal. We have suggested some broad interventions based on these similarities in inappropriate antibiotic use, and we have highlighted that stewardship cannot be achieved merely by increasing veterinary capacity or enforcing regulations. More practical solutions are needed to ensure access to the right antibiotics and to ensure this we need more rational guidelines for the different providers, access to some antibiotics for the paravets and better mentorship links between the formal and informal. We also need to bring the pharmaceutical industry on board with their support.  All this is highlighted in the discussion and in the conclusions (lines 721-725 pages 16 and lines 1092-1100 page 24)

We have also recommended further research on crossover use and its pathways and implications (lines 877-878, page 19)  

Point 7.

Reference should be in the same format. 

Response to point 7.

We have adjusted our reference list in line with MDPI’s guidance on how to format references for its journals as indicated here: https://mdpi-res.com/data/mdpi_references_guide_v5.pdf , following the American Society of Chemistry style as recommended.

Point 8.

The author could include the following references: 

(a) Sharma, G.; Mutua, F.; Deka, R.P.; Shome, R.; Bandyopadhyay, S.; Shome, B.; Goyal Kumar, N.; Grace, D.; Dey, T.K.; Venugopal, N.; et al. A qualitative study on antibiotic use and animal health management in smallholder dairy farms of four regions of India. Infect. Ecol. Epidemiol. 2020, 10, 1792033

(b) Durrance-Bagale A, Rudge JW, Singh NB, Belmain SR, Howard N. Drivers of zoonotic disease risk in the Indian subcontinent: A scoping review. One Health. 2021;13:100310. Published 2021 Aug 14. doi:10.1016/j.onehlt.2021.100310

(c) SwaiE. S., SchoonmanL., & DabornC. (2010). Knowledge and attitude towards zoonoses among animal health workers and livestock keepers in Arusha and Tanga, Tanzania. Tanzania Journal of Health Research, 12(4), 272-277. https://doi.org/10.4314/thrb.v12i4.54709 

(d) António Teixeira Rodrigues, Fátima Roque, Amílcar Falcão, Adolfo Figueiras, Maria Teresa Herdeiro Understanding physician antibiotic prescribing behaviour: a systematic review of qualitative studies, International Journal of Antimicrobial Agents, Volume 41, Issue 3 2013, 203-212, https://doi.org/10.1016/j.ijantimicag.2012.09.003. 

Response to point 8.

Thank you for these recommendations. We have reviewed these papers and added comments and references in the following places:

Line 713-715, page 16: We have added a sentence at lines relating to Sharma et al’s (2020) conclusion that the shortfall in veterinarians is likely to be the major reason for inadequate antibiotic use.

Line 746-749, page 17: We have cited Durrance-Bagale et al [2021] in as we echo their recommendation for a multistakeholder approach to developing interventions.

Line 746 (page 17) and Line 908 (page 20): We have cited Swai et al. (2010) in reference to the need for a one health, multi-stakeholder approach to interventions, and in relation to the need to raise awareness/educate livestock keepers.

Reviewer 3 Report

This MS describes a systematic analysis regarding the antibiotic use in smallholder livestock settings in rural West Bengal, India, which presented a good example of how misuse of the antibiotics promoted the emergence of ABU in India. The overall design of the experiment is elegant. However, several important details regarding the use of the antibiotics were missed. These clarity issues precluded the general readers to fully understand the virtual story.

Major
1) Introduction: 
line 58: the application of antibiotics in commercial farms should be mentioned before going to the details of smallholders.
2) Results:
line 173-180: The fact regarding the veterinary services and VPPs is largely descriptive. Author mentioned there is insufficient number of veterinarian. More speciefic data is needed, such as the number of veterinarian per farmer or he number of per population in the investigated regions.

line 229: the amount of antibiotics used in this regions should be elaborated.

line 284-314: the amount of antibiotics used for different purpose should be specified.

line 463: Authors stated that there is insufficient number of veterinary services in this region. More speciefic data is needed.

3) Discussion
line 706: crossover-use of human antibiotics in other countries should also be discussed.

Author Response

Point 1.  

Introduction: line 58: the application of antibiotics in commercial farms should be mentioned before going to the details of smallholders.

Response to point 1.

On the lines 64-67, page 2 we have made the recommended amendment and mentioned commercial farms before smallholders. We have also briefly elaborated on how antibiotics are used in commercial settings before discussing smallholder settings. In our discussion section, we have also included some discussion about commercial antibiotic usage (line 689-691), page 16

Point 2.

Results: line 173-180: The fact regarding the veterinary services and VPPs is largely descriptive. Author mentioned there is insufficient number of veterinarian. More specific data is needed, such as the number of veterinarian per farmer or the number of per population in the investigated regions. 

Response to point 2.

In the descriptive results regarding the veterinary services and VPPs, we have made the following amendments:

  • In India, the number of public veterinarians and other veterinary services is not publicly available. However, through personal communication with key contacts in the district, we have been able to gather reported data for the district of South 24 Parganas (the district of our study). There are 102 government sanctioned veterinary posts, but with 7 vacant posts. Therefore, there are 95 public veterinarians active in the district. Considering the district has a population of 8,161,961, there is approximately one public veterinarian per 85,915 members of the district’s population. This information is presented in our methods section on lines 985-990, page 21
  • In the descriptive results, we have added specific details regarding the provision of public veterinary services in both sites on lines 209-215. We have also provided these details in a table (table 3) on page 7.
  • In the methods section on lines 989-1001, pages 21-23, we have provided more specific geographical information regarding what “blocks” the two sites were located in along with a definition of what a block is (a rural subdivision of a district, subdivided for the purpose of local rural development).

Points 3 and 4.

Line 229: the amount of antibiotics used in this regions should be elaborated.

Line 284-314: the amount of antibiotics used for different purpose should be specified. 

Response to points 3 and 4.

Unfortunately, antibiotic usage surveillance data is not publicly available for India, nor for our regions. Our understanding of antibiotic consumption in food animals in India is limited to estimates of national antibiotic consumption by Van Boeckel et al. (2015) which we have presented in our introduction in the line 58-63, page 2. 

 Ref: Van Boeckel T.P. et. al. Global trends in antimicrobial use in food animals. PNAS. 2015: 112 (18); 5649 - 5654.  

Not being able to estimate the amount of antibiotics used according to species and purpose and assessing compliance to prescribed therapies were limitations of our study. However, the objective of our study was not to quantify antibiotic use but to find out the ‘how’ and ‘why’ of antibiotic use.  We aimed to identify and understand the perceptions, behaviours and practices surrounding ABU, including crossover-usage, through qualitative methods. We were able to identify the antibiotic classes used (and for what purposes) from the data collected during interviews, from the stocks of antibiotics maintained by the various antibiotic providers, and from medicines shown to us by livestock keepers reported to be used in livestock. However, quantifying the amounts of antibiotics used in our study settings is likely to be very challenging and would require a different study methodology and design. Even pharmaceutical sales data cannot provide precise information about household antibiotic use.  Livestock keepers were generally unaware of what antibiotics were, the nature of the drugs they used to treat their animals and rarely kept medicine packaging once a course of treatment was completed. Many of the providers did not keep medicine records, nor receipts of sale of medicines. Such factors would need consideration in the design of a study intending to quantify the amounts of antibiotics used by the different providers and by livestock keepers.  

Point 5.

line 463: Authors stated that there is insufficient number of veterinary services in this region. More specific data is needed. 

Response to point 5.

[Please See the response to point 2.] We have provided this additional information in lines 985-990, page 21

Point 6.

line 706: crossover-use of human antibiotics in other countries should also be discussed. 

Response to point 6.

From 852-858 on page 19, we have summarised the currently available literature pertaining to crossover-use of antibiotics. This issue has never been the focus of any studies and has only emerged as a set of incidental findings in a handful published studies. Thus, there is very limited evidence available on crossover use in the global literature. This is an under researched area and our study is one of the early ones to add to this increasing body of evidence.  The information described in these lines is the extent of what is available, to the best of our knowledge.   

Reviewer 4 Report

The manuscript titled "If it works in people, why not animals?’: A qualitative investigation of antibiotic use in smallholder livestock settings in rural West Bengal, India" described a survey about the use of human-antibiotics on animals livestock on a specific are in rural India. The manuscript is well written, however I have some criticisms. 

The introduction seems like script of a documentary about the situation of the development of resistance to antibiotics by livestock in that region of the world. I missed the scientific points in there. 

Neither in the introduction or discussion the authors explain the negative implications of the use of human-antibiotics for animals. Probably mentioning some examples must be enrich the discussion.

What do you mean with pluralistic rural health system? a short description must be included.

How the main problems could be address is another point that is poorly explain in the results and in the discussion. for example in the section 2.2.4.  How do you suggest that this situation must be corrected?

There are several statements that need some examples to be more accurate what do you mean. in the statement in line 606 starting in " These communities are particularly at....." 

The conclusions section do not provide any novel solution to the identified problems. 

Several question came to my mind for example. 

The guidelines that the authors suggest are for the veterinary doctors and paraprofessionals such as para-vets, livestock assistants and veterinary pharmacists? or for livestock keepers? 

Who do you think should be in charge of preparing these guidelines for the use of antibiotics in rural areas? The government through the ministry of agriculture and livestock? Do you think it should be supported by the international One Health WHO program?

Do you think that the guidelines will prevent the development of pathogenic microorganisms resistant to antibiotics? or will they only delay the emergence of antibiotic resistant strains? You mentioned in the introduction, the existence of resistance genes in nature, which suggests that the development of resistance to antibiotics is a natural phenomenon that may be inevitable.

Consider those question and include discussion in the introduction or discussion, after that the manuscript can be accepted

Author Response

Point 1.  

The introduction seems like script of a documentary about the situation of the development of resistance to antibiotics by livestock in that region of the world. I missed the scientific points in there. 

Response to point 1.

Our introduction was written with the intention of providing the context to the reader, to enumerate the current gaps in knowledge that exist in the use of antimicrobials in backyard production systems in low- and middle-income countries in order for the reader to better understand the scope of the study and the findings presented and discussed. It was taken into consideration that a reader may be unfamiliar with the rural Indian context. The scientific points in this section would relate to our current understanding of the drivers of antibiotic usage and resistance, the One Health implications of antibiotic resistance, and the evidence for inappropriate antibiotic usage in livestock in comparable smallholder settings.  

There is a paucity of social and behavioural data on antibiotic usage in India, and only a handful of studies have investigated antibiotic usage and resistance in smallholder settings in India and in low- and middle-income countries. We have limited understanding of the risks smallholder settings pose for the development and spread of antibiotic resistance. In the introduction section we have argued that to understand the AMR risks for smallholders, there is first a need to understand how antibiotics are used and/or misused, and the drivers of this misuse, in these contexts. We have now made efforts to reduce the length of the introduction and make our points more concise.  

Point 2.

Neither in the introduction or discussion the authors explain the negative implications of the use of human-antibiotics for animals. Probably mentioning some examples must be enrich the discussion. 

Response to point 2.

In the introduction, we have included a brief explanation of the negative implications of crossover-use from lines 127-131, page 3. We have addressed this in greater depth in our discussion from lines 862-878, page 19. Here we discuss the risks of antibiotic resistance developing against antibiotics of human significance, and the potential adverse effects, toxicity, or inefficacy of human antibiotics in livestock with select examples from the literature.

Crossover-use has not been widely reported, studied, or considered as a potential consequence of the misuse of human antibiotics in animals. The negative implications themselves have not been widely studied. Our discussion on the negative implications are interpreted from our current understanding of how antibiotic resistance develops and spreads when antibiotic formulations are used in their respective intended species (i.e., humans or animals). The adverse effects presented are from the scarce number of studies that are available to us. Furthermore, this was not a safety study to assess the impact of crossover over use in animal health and potential side effects that could have detrimental effects in animal health and welfare and productivity.

However, we have added (or made more explicit) how our understanding of the negative implications of crossover-use is limited by a lack of research into the practice, and further research is needed to evaluate these using appropriate study designs (lines 8720-878, page 19)

Point 3.

What do you mean with pluralistic rural health system? a short description must be included. 

Response to point 3.

We have now added a short explanation/definition of pluralistic health systems: “In India, farmers have access to a pluralistic healthcare system, whereby several systems of medicine are practiced through both formal and informal channels” on lines 94 to 96, page 2 where this terminology is mentioned with appropriate references for the definition included and added to our reference list as follows: 

  • Meessen B, Bigdeli M, Chheng K, Decoster K, Ir P, Men C, et al. Composition of pluralistic health systems: How much can we learn from household surveys? An exploration in Cambodia. Health Policy Plan. 2011. 
  • Bloom G, Standing H. Pluralism and marketisation in the health sector: meeting health needs in context of social change in low and middle-income countries. IDS Working Paper. 2001. 
  • Sudhinaraset M, Ingram M, Lofthouse HK, Montagu D. What Is the Role of Informal Healthcare Providers in Developing Countries? A Systematic Review. PLoS One. 2013. 

Point 4.

How the main problems could be address is another point that is poorly explain in the results and in the discussion. for example, in the section 2.2.4.  How do you suggest that this situation must be corrected? 

Response to point 4.

Thanks for pointing this out. We have now provided very clear, though broad, recommendations to many of the drivers of antibiotic usage and misuse in our discussion and conclusion sections. Please see the lines 737-764 on page 17, lines 882-940 on pages 19 and 20 and lines 1090-1107 on page 24.

Point 5.

There are several statements that need some examples to be more accurate what do you mean. In the statement in line 606 starting in " These communities are particularly at…" 

Response to point 5.

Thanks for this comment. Throughout our paper, we have identified statements where we could be more accurate and have made the necessary language amendments for this purpose. In regard to this particular statement, we have specified the term “these communities” to now read “humans and animals within these smallholder settings” (line 702-705)

Point 6.

The conclusions section do not provide any novel solution to the identified problems.  

Response to point 6.

We have changed the conclusion to highlight the novel findings and summarised the key recommendations for interventions.  A specific aim of our study was to identify the practice of crossover-use which we suspected was occurring in our sites as described in our introduction. We have now highlighted this novel finding for India in our conclusion on line 1086, page 24. Our key recommendations for interventions include the development of evidence-based guidelines for antibiotic usage in animal treatment as these do not exist at present. We have recommended harnessing all providers, professionals and para professionals as well as the current mentorship links between them. We have called for support from the pharmaceutical industry to address the shortcomings in the veterinary drugs and better policies to support access to some antibiotics through paraprofessionals.  

Point 7.

The guidelines that the authors suggest are for the veterinary doctors and paraprofessionals such as para-vets, livestock assistants and veterinary pharmacists? or for livestock keepers?  

Response to point 7.

On lines 737-740, page 17, we have now been more specific to explain that therapeutic antibiotic usage guidelines should be for both formal and informal animal healthcare practitioners, and drug shop owners (considering these often act as informal animal health providers). Considering the range of actors involved in the assessment of animal health and provision of antibiotics, we have specified that those responsible for guideline development take into consideration the diversity of animal healthcare providers in order to optimise antibiotic usage in these settings.

Point 8.

Who do you think should be in charge of preparing these guidelines for the use of antibiotics in rural areas? The government through the ministry of agriculture and livestock? Do you think it should be supported by the international One Health WHO program? 

Response to point 8.

Considering the diversity of actors involved in animal healthcare and provision of antibiotics in these settings, we have highlighted the need for a multi-stakeholder approach for developing a multi-sectoral intervention. Our ongoing stakeholder consultations (part of this project) have recommended that we need to bring together local governmental and non-governmental stakeholders for guideline development, feasibility testing and implementation. Epidemiological assessments on antibiotic sensitivity patterns and the views of local professional and paraprofessional providers must inform and guide the formulation of guidelines, for these to be effective (lines 759-764).

Point 9.

Do you think that the guidelines will prevent the development of pathogenic microorganisms resistant to antibiotics? or will they only delay the emergence of antibiotic resistant strains? You mentioned in the introduction, the existence of resistance genes in nature, which suggests that the development of resistance to antibiotics is a natural phenomenon that may be inevitable. 

Response to point 9.

We agree that the development of guidelines in isolation won’t be sufficient to prevent the development of resistance. A lot would depend on how they are implemented and complied with by healthcare providers.  However, if they could successfully reduce or optimise antibiotic, they could significantly slow the process of resistance developing in our study setting. It has been seen in studies in hospitals and in some high-income outpatient settings that reductions in antibiotic usage have lowered resistance levels. However, for guidelines to be complied with and to lead to optimal antibiotic use, the other social and economic drivers will also need to be addressed.

The reference to antibiotic resistance genes in the introduction was to highlight that resistance can transfer between the human, animal, and environmental domains. It is a fact that antimicrobial resistance occurs naturally amongst certain bacteria and other microorganisms; however, the use of antibiotics can exacerbate and speed up the acquisition of antimicrobial resistance in bacterial populations. In fact, it is acquired antimicrobial resistance particularly that acquired through horizontal transfer of plasmids and other mobile structures that is considered to be the most concerning from a public health perspective.

Round 2

Reviewer 1 Report

Dear Authors,

I read with pleasure the new version of your manuscript and the responses to my previous comments.  After understanding your primary aim and the qualitative analysis done, I believe that the manuscript can be considered for publication.  As I wrote, this work is very important for both local and global health, since nowadays the risk of MDR bacteria is a well-known global problem.

Reviewer 4 Report

The manuscript now is suitable to publish